# Comparing the Effects of Canagliflozin vs. Glimepiride by Body Mass Index in Patients with Type 2 Diabetes and Chronic Heart Failure: A Subanalysis of the CANDLE Trial

**DOI:** 10.3390/biomedicines10071656

**Published:** 2022-07-09

**Authors:** Akira Sezai, Atsushi Tanaka, Takumi Imai, Keisuke Kida, Hisakuni Sekino, Toyoaki Murohara, Masataka Sata, Norio Suzuki, Koichi Node

**Affiliations:** 1Department of Cardiovascular Surgery, Nihon University School of Medicine, Tokyo 173-8610, Japan; 2Department of Cardiovascular Medicine, Saga University, Saga 849-8501, Japan; tanakaa2@cc.saga-u.ac.jp (A.T.); node@cc.saga-u.ac.jp (K.N.); 3Department of Medical Statistics, Osaka Metropolitan University Graduate School of Medicine, Osaka 530-0001, Japan; takumi.imai@omu.ac.jp; 4Department of Pharmacology, St. Marianna University School of Medicine, Kawasaki 216-8511, Japan; heart-kida@marianna-u.ac.jp; 5Sekino Hospital, Tokyo 171-0014, Japan; sekinoh@sekino-hospital.com; 6Department of Cardiology, Nagoya University School of Medicine, Nagoya 466-8550, Japan; murohara@med.nagoya-u.ac.jp; 7Department of Cardiovascular Medicine, Tokushima University Graduate School of Biomedical Science, Tokushima 770-8503, Japan; masataka.sata@tokushima-u.ac.jp; 8Division of Cardiology, Department of Internal Medicine, St. Marianna University School of Medicine, Kawasaki 216-8511, Japan; n2suzuki@marianna-u.ac.jp

**Keywords:** sodium–glucose transporter 2 inhibitors, diabetes, heart failure, NT-proBNP, BMI

## Abstract

Background: We present results of a 24-week comparative study of the effects of the sodium–glucose cotransporter 2 (SGLT2) inhibitor canagliflozin vs. the sulfonylurea glimepiride, by baseline body mass index (BMI), in patients with type 2 diabetes and chronic heart failure. Methods: We conducted a post hoc analysis of the CANDLE trial. This subanalysis evaluated NT-proBNP, BMI, and other laboratory parameters, according to the subgroups stratified by BMI ≥ 25 kg/m^2^ vs. BMI < 25 kg/m^2^. Results: A group ratio of proportional changes in the geometric means of NT-proBNP was 0.99 (*p* = 0.940) for the subgroup with BMI ≥ 25 kg/m^2^ and 0.85 (*p* = 0.075) for the subgroup with BMI < 25 kg/m^2^, respectively. When baseline BMI was modeled as a continuous variable, results for patients with BMI < 30 kg/m^2^ showed a slightly smaller increase in NT-proBNP in the canagliflozin group vs. the glimepiride group (*p* = 0.295); that difference was not seen among patients with BMI ≥30 kg/m^2^ (*p* = 0.948). Irrespective of obesity, the canagliflozin group was associated with significant reduction in BMI compared to the glimepiride group. Conclusion: There was no significant difference in the effects of canagliflozin, relative to glimepiride, on NT-proBNP concentrations irrespective of baseline obesity. UMIN clinical trial registration number: UMIN000017669.

## 1. Introduction

Meta-analyses involving large-scale studies of patients with diabetes have indicated that the use of sodium–glucose cotransporter 2 (SGLT2) inhibitors is associated with a decreased risk of three-point major adverse cardiovascular events, including cardiovascular death [1,2,3,4]. These cardioprotective effects of SGLT2 inhibitors have attracted attention among cardiologists. More recently, several large clinical studies evaluating heart failure with reduced ejection fraction (HFrEF) have been conducted [5,6]. In 2021, another large-scale clinical study of heart failure with preserved ejection fraction (HFpEF) demonstrated reduction in cardiovascular death, heart failure, and hospitalization [7]. Consequently, SGLT2 inhibitors have become established as a new class of drugs for preventing heart failure (HF).

The CANDLE trial (UMIN000017669, http://www.umin.ac.jp/ (accessed on 14 June 2022)) is an investigator-initiated, multicenter, open-label, randomized controlled trial comparing the effects of canagliflozin and glimepiride in Japanese patients with type 2 diabetes (T2D) and chronic heart failure (chronic HF) [8,9]. In the CANDLE trial, the primary endpoint was the percentage change from baseline in N-terminal pro-brain natriuretic peptide (NT-proBNP) at 24 weeks, and the secondary endpoints were vital signs (body weight, blood pressure, and heart rate), glycemic control (HbA1c, fasting plasma glucose), estimated plasma volume calculated by the Strauss formula, echocardiographic measures, NYHA functional classification, and chronic heart-failure-related quality of life evaluated by scaled responses to the Minnesota living with heart failure questionnaire [8,9]. Results from this trial indicate canagliflozin administered for 24 weeks to elderly patients with diabetes and stable chronic HF was not noninferior regarding percentage change in N-terminal pro-brain natriuretic peptide (NT-proBNP) levels, possibly due to the large variation in NT-proBNP levels. Secondly, the trial results demonstrate that absolute reduction in NT-proBNP, a secondary endpoint, was greater in the canagliflozin group vs. the glimepiride group, especially among patients with elevated NT-proBNP. Thirdly, in a subgroup of patients with HFpEF, use of canagliflozin reduced NT-proBNP levels and improved New York Heart Association (NYHA) functional classification more than use of glimepiride did [9]. Lastly, results from an additional subanalysis of the trial involving patients with lower left ventricular diastolic function did not show a statistically significant difference in NT-proBNP among those who received canagliflozin compared with those who received glimepiride [10].

Findings from a subanalysis of a large international study demonstrated similar efficacy of SGLT2 inhibitors for reducing body mass index (BMI) among patients with minor obesity in Asian, European, and American populations [11,12]. We conducted the subanalysis described herein to clarify whether the effects of canagliflozin on NT-proBNP, BMI, and other laboratory parameters differ among patients with vs. without obesity.

## 2. Materials and Methods

### 2.1. Study Design and Participants

This report was a post hoc subanalysis of the CANDLE trial (UMIN000017669), an investigator-initiated, multicenter, prospective, randomized, open-label clinical trial primarily to assess the effect of 24 weeks of add-on canagliflozin treatment, versus glimepiride, on NT-proBNP concentration in patients with T2D and concomitant chronic HF [8,9]. The details of the original study design and participants criteria have been reported previously [8,9]. Briefly, eligibilities were adults with T2D and chronic HF excluding New York Heart Association (NYHA) class IV and clinically stable four weeks prior to study enrollment. Key exclusion criteria were severe renal dysfunction (estimated glomerular filtration rate (eGFR) < 45 mL/min/1.73 m^2^ or on dialysis), patients with malnutrition, patients in perioperative period around screening visit, patients with severe infection or trauma at trial screening, patients with a malignancy, and a recent history of cardiovascular disease (CVD) within 3 months prior to screening. Using a web-based minimization method balanced for age (<65, ≥65 yr), HbA1c level (<6.5%, ≥6.5%), and left ventricular ejection fraction (LVEF; <40%, ≥40%) at the time of screening, eligible participants were assigned randomly to either canagliflozin (100 mg daily) or glimepiride (starting dose 0.5 mg daily) treatment groups. All participants received the study treatment for 24 weeks and were managed based on the local guidelines for T2D and chronic HF. In the glimepiride group, increases in the dose of glimepiride were allowed according to the individual’s glycemic control and investigator’s judgment. The participant’s background medications were, in principle and if possible, maintained during the study interval within clinically permissible range.

The trial was approved by the institutional review boards of the individual sites and conducted in accordance with the Declaration of Helsinki. All participants provided written, informed consent prior to screening and randomization.

In this study, we used the obesity criteria by the World Health Organization (WHO), and obesity was defined as ≥BMI 25 kg/m^2^ while <25 kg/m^2^ as non-obesity. If the baseline BMI value was missing, the participant was excluded from the analyses.

### 2.2. Measurements and Endpoints

The details of the original outcome measures in the CANDLE trial have been described previously [8,9]. In brief, vital signs and blood samples were collected at baseline, week 4, week 12, and week 24, and routine laboratory parameters were measured at each local site. NT-proBNP concentrations were assessed at each local site and measured in a blinded manner at a central core laboratory (SRL, Inc. Tokyo, Japan) using an electrochemiluminescence immunoassay (Roche, Basel, Switzerland). In this study, changes in NT-proBNP, BMI, hemoglobin, hematocrit, eGFR and hemoglobin A1c (HbA1c) from baseline to each visit were targets of analyses.

### 2.3. Statistical Analysis

All statistical analyses were performed according to the intention-to-treat principle. The baseline characteristics were expressed using frequencies with percentages for categorical variables and means with standard deviations or median with interquartile range for continuous variables. For NT-proBNP, the analysis was performed using linear models on the logarithmic scale with adjustment for the baseline value, for the subgroups of baseline BMI < 25 kg/m^2^ and ≥25 kg/m^2^. Proportional changes in the geometric means from baseline to 24 weeks for both treatment groups and the ratio of the proportional changes between treatment groups were estimated with 95% confidence intervals (CIs) and compared among baseline BMI subgroups. In addition, another statistical analysis with baseline continuous BMI value included in the model of NT-proBNP using the restricted cubic spline function was performed. The estimated proportional changes in the geometric means for both treatment groups were plotted against baseline continuous BMI value. For evaluation of the follow-up BMI, hemoglobin, hematocrit, eGFR and HbA1c, the analyses were performed using linear mixed models with adjustment for the baseline value, for the subgroups of baseline BMI < 25 kg/m^2^ and ≥25 kg/m^2^. The mean values for both treatment groups and treatment group differences at 4, 12, and 24 weeks were estimated with 95% CIs, and compared among baseline BMI subgroups. All analyses were conducted using R, Version 4.1.0 (R Foundation for Statistical Computing, Vienna, Austria) at a two-sided significance level of 0.05. No adjustment for multiplicity was considered in the post hoc subanalysis.

## 3. Results

### 3.1. Baseline Clinical Characteristics

This subanalysis of the CANDLE study involved 111 patients with baseline BMI < 25 kg/m^2^, of whom 55 patients were assigned to receive canagliflozin and 56 were assigned to receive glimepiride, and 121 patients with baseline BMI ≥ 25 kg/m^2^, including 58 patients assigned to the canagliflozin group and 63 assigned to the glimepiride group (Figure 1).

Table 1 shows the baseline demographics and clinical characteristics for the full analysis set, stratified by baseline BMI. Causes of heart failure were less likely to involve ischemia, valvular disease, or arrhythmia and more likely to include hypertension or dilated cardiomyopathy among patients with baseline BMI ≥ 25 kg/m^2^ compared with those with BMI < 25 kg/m^2^. Overall, the baseline NT-proBNP level in patients with baseline BMI < 25 kg/m^2^ (median 330.0 pg/mL [interquartile range 149.0–719.0 pg/mL]) was higher than those with baseline BMI ≥ 25 kg/m^2^ (median 205.0 pg/mL [interquartile range 69.5–458.0 pg/mL], *p* = 0.004 by Wilcoxon rank sum test). The median NT-proBNP levels in each treatment group are shown in Table 1. Detailed information on background medications for T2D and chronic HF have been previously shown elsewhere [9,13].

### 3.2. Effects of Treatment

#### 3.2.1. NT-proBNP

Among patients with baseline BMI < 25 kg/m^2^, levels of NT-proBNP were not different before vs. after (24 weeks) treatment in both the canagliflozin group and the glimepiride group; proportional changes in geometric means were 0.95 (95% CI 0.81 to 1.07) and 1.07 (95% CI 0.92 to 1.25), respectively. The group ratio of the proportional change in geometric means was 0.85 (95% CI 0.75 to 1.02, *p* = 0.075: Figure 2a). Similarly, for patients with baseline BMI ≥ 25 kg/m^2^, the levels of NT-proBNP were not different before vs. after (24 weeks) treatment in both the canagliflozin group and the glimepiride group; proportional changes in geometric means were 1.01 (95% CI 0.89 to 1.14) and 1.05 (95% CI 0.93 to 1.17), respectively. The group ratio of the proportional change in geometric means was 0.99 (95% CI 0.84 to 1.18, *p* = 0.940: Figure 2b). Although the group ratio for patients with baseline BMI < 25 kg/m^2^ was smaller than that of patients with baseline BMI ≥ 25 kg/m^2^, there was no statistical difference by baseline BMI subgroups (*p* = 0.222). When the proportional changes in NT-proBNP geometric means were modeled by continuous baseline BMI values, the magnitude of proportional change in canagliflozin group was lower than that of the glimepiride group in BMI < 30 kg/m^2^ approximately (Figure 2c).

#### 3.2.2. BMI

Post-treatment reductions in BMI were observed in the canagliflozin group but not in the glimepiride group for both baseline BMI subgroups (Figure 3a,b). Although the reduction in BMI due to canagliflozin compared by glimepiride was larger in patients with baseline BMI ≥ 25 kg/m^2^, there was no statistical difference in baseline BMI subgroups in 24 weeks (*p* = 0.083).

#### 3.2.3. Hemoglobin and Hematocrit

Post-treatment increases in hemoglobin and hematocrit levels were observed in the canagliflozin group but not in the glimepiride group for both baseline BMI subgroups (Table 2 and Figure 3c,d). Although the increase in hemoglobin due to canagliflozin compared by glimepiride was larger in patients with baseline BMI ≥ 25 kg/m^2^, there was no statistical difference in baseline BMI subgroups (*p* = 0.136). On the other hand, the increase in hematocrit due to canagliflozin compared with glimepiride was statistically larger in patients with baseline BMI ≥ 25 kg/m^2^ (*p* = 0.037).

#### 3.2.4. eGFR

For both baseline BMI subgroups, the post-treatment reduction in eGFR due to canagliflozin was greater than that of glimepiride over 24 weeks, but the reduction became milder at 12 and 24 weeks (Table 2). There was no statistical difference in treatment effects for eGFR in baseline BMI subgroups over 24 weeks.

#### 3.2.5. Hemoglobin A1c

While a post-treatment reduction in HbA1c due to glimepiride was observed, the mean HbA1c level was unchanged in the canagliflozin group for both the baseline BMI subgroups (Table 2). Although the difference in changes in HbA1c at 12 and 24 weeks between canagliflozin and glimepiride was larger in patients with baseline BMI < 25 kg/m^2^, there was no statistical difference in baseline BMI subgroups.

## 4. Discussion

This subanalysis did not demonstrate a statistically significant proportional change in NT-proBNP level after treatment for both treatment groups and baseline BMI subgroups. However, in the subgroup with baseline BMI < 25 kg/m^2^, the canagliflozin group had a trend of reduced NT-pro BNP level compared with glimepiride. This may suggest the varying effects on NT-proBNP depending on baseline BMI. Based on the result for proportional change in NT-proBNP geometric means modeled by continuous baseline BMI values, a lower proportional change for canagliflozin was observed in baseline BMI < 30 kg/m^2^. The result is very intriguing, and it was suggested that a new finding may be obtained by increasing the number of patients with BMI ≥ 30 kg/m^2^ (obesity, II and III).

Fedele et al. reported a nearly 2-fold increase in NT-proBNP levels after a brief period of lifestyle intervention among normotensive patients with severe obesity and without cardiac disease [14]. Levels of NT-proBNP generally are lower in patients with obesity. Indeed, baseline values in our subanalysis showed that NT-proBNP levels were lower among patients with BMI ≥ 25 kg/m^2^ than among those with BMI < 25 kg/m^2^. Furthermore, it has been suggested that changes in NT-proBNP differ between patients with vs. without obesity. In the DEFINE-HF trial, in which patients with HFrEF received dapagliflozin for 12 weeks, treatment did not affect mean NT-proBNP levels but increased the proportion of patients who experienced a clinically meaningful improvement in their cardiovascular health status or natriuretic peptide levels [15]. Jensen et al. reported that NT-proBNP levels did not decline after 12 weeks of treatment with empagliflozin [16]. Ferrannini et al. also reported that when empagliflozin was administered for 4 weeks to patients with type 2 diabetes, NT-proBNP levels did not change but plasma erythropoietin concentrations increased by 31% [17]. It is reported that NT-proBNP concentration is higher if a patient is lean while it is lower in an obese patient. There are many reports about the reduction in body weight and ventricular load by the SGLT2 inhibitor. In this subanalysis, the canagliflozin group resulted in a decrease in BMI, but NT-proBNP was not different between both drug groups and between before and after treatment with canagliflozin. One of the possible reasons behind this phenomenon appears to be a decrease in body weight by canagliflozin and a resultant offsetting of the NT-proBNP reduction mediated by the canagliflozin-induced cardiac unload. Since SGLT2 inhibitor deceases body weight, NT-proBNP might have been an inappropriate biomarker to directly reflect the reduction in cardiac load via SGLT2 inhibition. However, there are still unresolved aspects behind this phenomenon, and there is room for discussions [18]. It is therefore necessary to search for more appropriate biomarkers for efficacy assessment and monitoring of cardiovascular effects by SGLT2 inhibitor.

Immediately after patients receive an SGLT2 inhibitor, hematocrit levels increase due to dehydration; this transient increase in hematocrit is a result of increased erythropoietin [19]. However, our study did not measure erythropoietin concentration, and this is hypothetical. We consider that it is warranted to measure hematocrit level and erythropoietin for an extended period.

The CANDLE trial was a 24-week study. Subsequently, it was considered necessary to evaluate NT-proBNP levels for a longer period after treatment with an SGLT2 inhibitor. In this subanalysis, more than 70% of patients with baseline BMI ≥ 25 kg/m^2^ had HFpEF. Results may have been affected by a large proportion of patients with lower baseline NT-proBNP levels. Additionally, levels of adiponectin may have influenced NT-proBNP levels in patients with obesity. Levels of NT-proBNP have been shown to be inversely related to metabolic syndrome and obesity, and adiposity profile factors such as lipolysis and fat mobilization affect NT-proBNP levels [20,21]. Garvey et al. demonstrated that compared with glimepiride, canagliflozin significantly decreased levels of serum leptin and increased levels of serum adiponectin. It has been reported that treatment with canagliflozin improves adipose tissue functions; modifies levels of serum leptin, adiponectin, and interleukin 6; and may favorably affect insulin sensitivity and other risk factors for cardiovascular disease [22]. SGLT2 inhibitors have been shown to reduce body weight. Although this subanalysis did not demonstrate a statistically significant association between BMI and NT-proBNP levels, this potential relationship warrants further investigation. SGLT2 inhibitor-induced reductions in body weight likely are associated with subsequent decreases in other parameters such as interstitial fluid and fat mass [23]. Previously, we demonstrated by computed tomography that patients treated with canagliflozin experienced marked reductions in visceral fat and subcutaneous fat that were strongly correlated with decreased BMI [24].

Sakai et al. administered luseogliflozin to Japanese patients with type 2 diabetes and investigated the effects, stratified by baseline BMI. Decreases in HbA1c were observed regardless of baseline BMIs. While reductions in body weight were most pronounced among patients with higher BMIs, similar effects were observed in patients without obesity [25]. Our subanalysis demonstrated that, irrespective of BMI at baseline, BMI declined significantly after treatment in the canagliflozin group but not in the glimepiride group. However, treatment with canagliflozin was not associated with decreased levels of HbA1c, and this was consistent regardless of baseline obesity. In the CANTATA-SU trial, which compared canagliflozin 100 mg and 300 mg vs. glimepiride, the results at 52 weeks indicated treatment with canagliflozin 300 mg significantly decreased HbA1c, but canagliflozin 100 mg was not significantly different compared with glimepiride [26]. In comparison, the CANDLE trial used a low dose (100 mg) of canagliflozin and a shorter observation period, which may explain differences between the results from these studies.

Conventional diabetes drugs exert their effects by promoting insulin secretion or improving insulin resistance. Sulfonylurea drugs promote insulin secretion, thereby controlling blood glucose. In contrast, SGLT2 inhibitors decrease blood glucose not by affecting insulin directly but by increasing excretion of urinary glucose, thereby indirectly inhibiting insulin secretion [27]. SGLT2 inhibitors also are thought to protect pancreatic beta cells [28]. Results of this subanalysis demonstrated the characteristic effects of sulfonylurea drugs and SGLT2 inhibitors, regardless of baseline BMI.

## 5. Limitation

This subanalysis has several limitations. First of all, a shorter observation period (24 weeks) and a small sample size might have affected outcomes in the present analyses. In particular, we could have a subpopulation only in patients with BMI ≥ 25 kg/m^2^ and those with BMI < 25 kg/m^2^. If the number of patients with BMI > 30 kg/m^2^ was larger, the result might have significantly been altered. Additionally, one of the inclusion criteria of this study was BMI > 18.5 kg/m^2^ and patients at low body weight were excluded. A comparison between BMI < 18.5 kg/m^2^ (low body weight), BMI ≥ 18.5 kg/m^2^, <25 kg/m^2^ (standard body weight), BMI ≥ 25 kg/m^2^, <30 kg/m^2^ (obesity I) and BMI ≥ 30 kg/m^2^ (obesity, II and III) may bring about new findings.

## 6. Conclusions

In this subanalysis from the CANDLE trial, there was no significant difference in the effects of 24-week canagliflozin treatment, relative to glimepiride, on NT-proBNP concentrations, irrespective of baseline obesity. Further studies are needed to assess whether the cardiovascular effects of SGLT2 inhibitor differ among obesity categories.

## Figures and Tables

**Figure 1 biomedicines-10-01656-f001:**
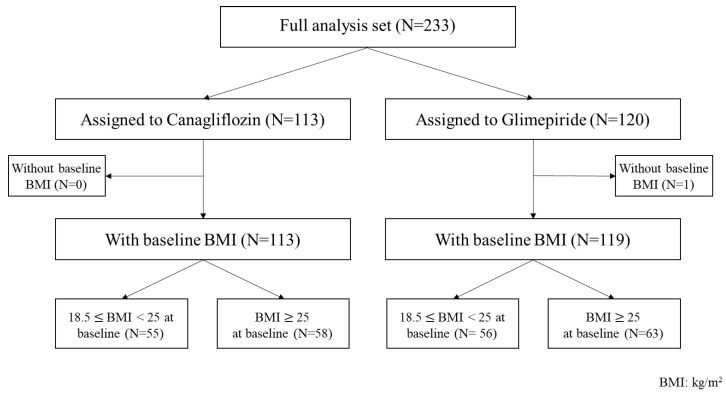
Flow-chart of this study. BMI, body mass index.

**Figure 2 biomedicines-10-01656-f002:**
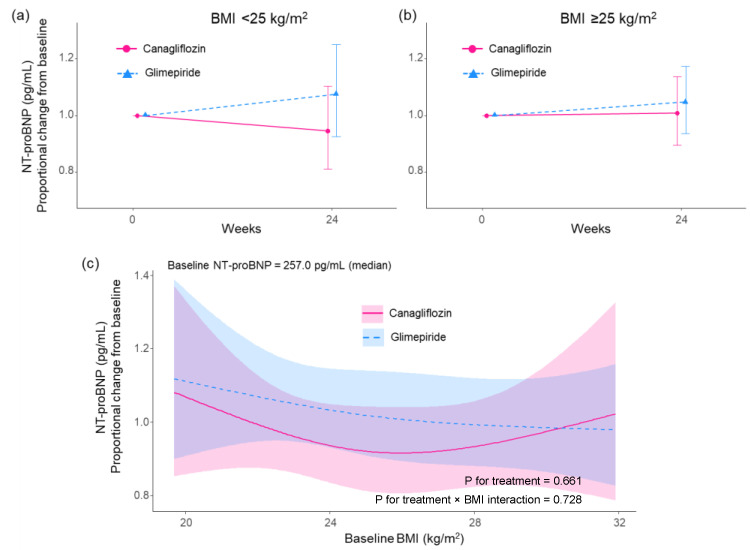
Proportional changes in N-terminal pro-brain natriuretic peptide (NT-proBNP) from baseline to week 24 in the subgroups stratified by the baseline body mass index (BMI). The data are expressed as estimate and 95% confidence interval. (**a**) Proportional changes in NT-proBNP geometric means from baseline to week 24 for the patients with baseline BMI < 25 kg/m^2^. (**b**) Proportional changes in NT-proBNP geometric means from baseline to week 24 for the patients with baseline BMI ≥ 25 kg/m^2^. (**c**) Proportional change in NT-proBNP geometric means modeled by continuous baseline BMI values using the restricted cubic spline function. Lines and light-colored areas represent pointwise estimates and 95% confidence intervals.

**Figure 3 biomedicines-10-01656-f003:**
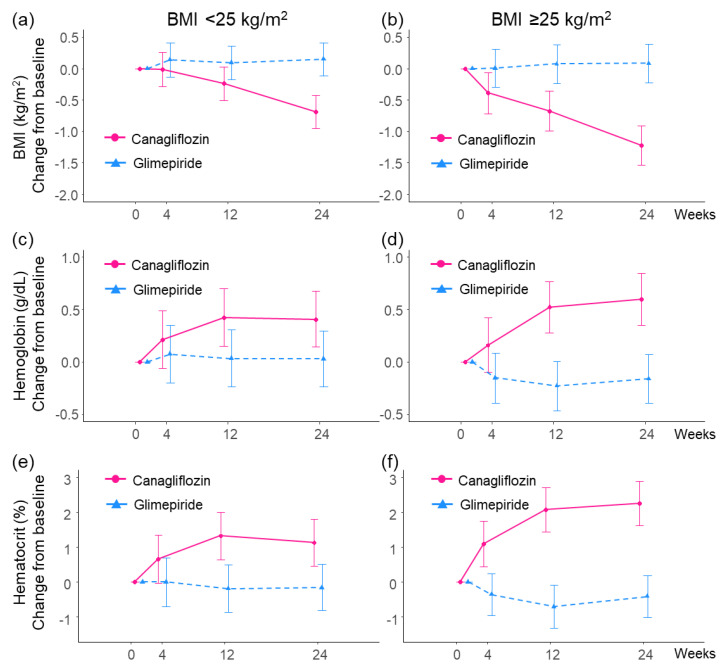
Change in mean body mass index (BMI), hemoglobin and hematocrit from baseline to 4, 12 and 24 weeks in the subgroups stratified by the baseline BMI. The data are expressed as estimate an 95% confidence interval. (**a**) Change in mean BMI from baseline to 4, 12 and 24 weeks for the patients with baseline BMI < 25 kg/m^2^. (**b**) Change in mean BMI from baseline to 4, 12 and 24 weeks for the patients with baseline BMI ≥ 25 kg/m^2^. (**c**) Change in mean hemoglobin from baseline to 4, 12 and 24 weeks for the patients with baseline BMI < 25 kg/m^2^. (**d**) Change in mean hemoglobin from baseline to 4, 12 and 24 weeks for the patients with baseline BMI ≥ 25 kg/m^2^. (**e**) Change in mean hematocrit from baseline to 4, 12 and 24 weeks for the patients with baseline BMI < 25 kg/m^2^. (**f**) Change in mean hematocrit from baseline to 4, 12 and 24 weeks for the patients with baseline BMI ≥ 25 kg/m^2^.

**Table 1 biomedicines-10-01656-t001:** Baseline characteristics of the patients.

	Patients BMI < 25 kg/m^2^ at Baseline (N = 111)		Patients BMI ≥ 25 kg/m^2^ at Baseline (N = 121)	
Characteristic	Canagliflozin Group (N = 55)	Glimepiride Group (N = 56)	Canagliflozin Group (N = 58)	Glimepiride Group (N = 63)
Age (mean ± SD), years	70.8 ± 8.6	72.1 ± 7.3	65.9 ± 10.3	66.1 ± 11.9
Sex, no. (%)				
Male	41 (74.5%)	37 (66.1%)	47 (81.0%)	48 (76.2%)
Female	14 (25.5%)	19 (33.9%)	11 (19.0%)	15 (23.8%)
NT-proBNP (median [IQR]), pg/mL	285.0 [116.0 to 662.0]	374.5 [153.8 to 774.0]	213.0 [112.0 to 412.0]	158.0 [58.2 to 499.2]
BMI (mean ± SD), kg/m^2^	22.4 ± 1.6	22.4 ± 1.8	28.0 ± 2.8	28.6 ± 3.6
Hemoglobin (mean ± SD), g/dL	13.4 ± 1.8	13.4 ± 1.7	14.1 ± 1.6	14.1 ± 1.7
Hematocrit (mean ± SD), %	40.8 ± 5.1	40.6 ± 4.0	42.4 ± 4.5	42.1 ± 4.8
eGFR (mean ± SD), mL/min/1.73 m^2^	62.9 ± 14.8	61.6 ± 12.6	65.2 ± 15.7	64.8 ± 16.7
HbA1c (mean ± SD), %	6.9 ± 0.8	7.0 ± 1.0	7.0 ± 0.7	7.1 ± 0.9
LVEF < 50%, no. (%)	19 (34.5%)	15 (26.8%)	15 (26.3%)	18 (28.6%)
Heart failure cause, no. (%)				
Ischemia	29 (52.7%)	22 (39.3%)	25 (43.1%)	24 (38.1%)
Hypertension	11 (20.0%)	13 (23.2%)	21 (36.2%)	17 (27.0%)
Valvular disease	14 (25.5%)	11 (19.6%)	5 (8.6%)	6 (9.5%)
Dilated cardiomyopathy	6 (10.9%)	8 (14.3%)	11 (19.0%)	11 (17.5%)
Arrhythmia	15 (27.3%)	20 (35.7%)	14 (24.1%)	12 (19.0%)

Abbreviations, SD, standard deviation; BMI, body mass index; NT-proBNP, N-terminal pro-brain natriuretic peptide; IQR, interquartile range; eGFR, estimated glomerular filtration rate; HbA1c, Hemoglobin A1c; LVEF, left ventricular ejection fraction.

**Table 2 biomedicines-10-01656-t002:** Changes from baseline to week 24 in BMI, Hemoglobin, Hematocrit, eGFR, and HbA1c.

		Patients BMI < 25 kg/m^2^ at Baseline (N = 111)			Patients BMI ≥ 25 kg/m^2^ at Baseline (N = 121)			
		Change from Baselinea		Group Difference ^b^	Change from Baseline ^a^		Group Difference ^b^	
Variables	Visit	Canagliflozin (N = 55)	Glimepiride (N = 56)	Canagliflozin vs. Glimepiride	Canagliflozin (N = 58)	Glimepiride (N = 63)	Canagliflozin vs. Glimepiride	*p* for int.
Body mass index (kg/m^2^)	4 weeks	−0.01 (−0.28 to 0.26)	0.14 (−0.13 to 0.41)	−0.15 (−0.58 to 0.28) *p* = 0.495	−0.38 (−0.71 to −0.06)	0.00 (−0.30 to 0.31)	−0.43 (−0.84 to −0.02) *p* = 0.038	0.353
	12 weeks	−0.24 (−0.51 to 0.03)	0.09 (−0.17 to 0.36)	−0.33 (−0.75 to 0.09) *p* = 0.125	−0.67 (−0.99 to −0.36)	0.07 (−0.23 to 0.38)	−0.79 (−1.19 to −0.38) *p* < 0.001	0.127
	24 weeks	−0.69 (−0.95 to −0.43)	0.16 (−0.11 to 0.42)	−0.84 (−1.25 to −0.42) *p* < 0.001	−1.22 (−1.54 to −0.91)	0.08 (−0.22 to 0.39)	−1.35 (−1.75 to −0.95) *p* < 0.001	0.083
Hemoglobin (g/dL)	4 weeks	0.21 (−0.06 to 0.49)	0.07 (−0.20 to 0.35)	0.18 (−0.18 to 0.54) *p* = 0.327	0.16 (−0.10 to 0.42)	−0.15 (−0.39 to 0.09)	0.31 (−0.04 to 0.66) *p* = 0.082	0.604
	12 weeks	0.42 (0.15 to 0.70)	0.03 (−0.24 to 0.30)	0.42 (0.07 to 0.77) *p* = 0.020	0.52 (0.28 to 0.77)	−0.23 (−0.47 to 0.01)	0.75 (0.41 to 1.09) *p* < 0.001	0.189
	24 weeks	0.41 (0.14 to 0.68)	0.03 (−0.24 to 0.30)	0.39 (0.04 to 0.74) *p* = 0.027	0.60 (0.35 to 0.84)	−0.16 (−0.40 to 0.07)	0.76 (0.42 to 1.10) *p* < 0.001	0.136
Hematocrit (%)	4 weeks	0.65 (−0.03 to 1.34)	0.00 (−0.69 to 0.69)	0.70 (−0.23 to 1.63) *p* = 0.140	1.09 (0.44 to 1.74)	−0.36 (−0.97 to 0.25)	1.45 (0.54 to 2.35) *p* = 0.002	0.263
	12 weeks	1.32 (0.63 to 2.00)	−0.19 (−0.86 to 0.49)	1.55 (0.63 to 2.47) *p* = 0.001	2.07 (1.44 to 2.71)	−0.70 (−1.31 to −0.09)	2.77 (1.88 to 3.67) *p* < 0.001	0.061
	24 weeks	1.14 (0.47 to 1.80)	−0.16 (−0.82 to 0.51)	1.33 (0.42 to 2.23) *p* = 0.004	2.25 (1.62 to 2.89)	−0.41 (−1.01 to 0.19)	2.67 (1.78 to 3.55) *p* < 0.001	0.037
eGFR (mL/min/1.73 m^2^)	4 weeks	−4.42 (−6.28 to −2.56)	0.89 (−0.96 to 2.74)	−5.20 (−7.73 to −2.66) *p* < 0.001	−3.83 (−5.69 to −1.97)	−0.51 (−2.23 to 1.22)	−3.27 (−5.72 to −0.82) *p* = 0.009	0.284
	12 weeks	−3.37 (−5.21 to −1.53)	−1.28 (−3.11 to 0.55)	−1.99 (−4.50 to 0.51) *p* = 0.119	−3.60 (−5.40 to −1.80)	−1.23 (−2.96 to 0.50)	−2.34 (−4.76 to 0.07) *p* = 0.057	0.842
	24 weeks	−2.86 (−4.68 to −1.05)	−2.12 (−3.92 to −0.31)	−0.67 (−3.14 to 1.80) *p* = 0.596	−2.99 (−4.78 to −1.20)	−1.21 (−2.94 to 0.52)	−1.79 (−4.19 to 0.61) *p* = 0.144	0.524
HbA1c (%)	4 weeks	0.11 (−0.05 to 0.26)	−0.08 (−0.24 to 0.07)	0.15 (−0.06 to 0.36) *p* = 0.156	0.09 (−0.07 to 0.26)	−0.11 (−0.26 to 0.05)	0.16 (−0.05 to 0.36) *p* = 0.129	0.976
	12 weeks	0.11 (−0.04 to 0.26)	−0.33 (−0.49 to −0.18)	0.41 (0.20 to 0.61) *p* < 0.001	0.05 (−0.11 to 0.22)	−0.20 (−0.36 to −0.04)	0.22 (0.02 to 0.42) *p* = 0.034	0.205
	24 weeks	0.00 (−0.16 to 0.15)	−0.47 (−0.62 to −0.32)	0.43 (0.22 to 0.63) *p* < 0.001	0.01 (−0.15 to 0.18)	−0.21 (−0.36 to −0.05)	0.18 (−0.02 to 0.38) *p* = 0.077	0.098

Abbreviations: BMI, body mass index; eGFR, estimated glomerular filtration rate; HbA1c, Hemoglobin A1c; P for int., *p* value for interaction between treatment and baseline BMI ^a^ Estimate of mean change from baseline and 95% confidence interval ^b^ Estimate of difference in mean change from baseline between treatment groups, 95% confidence interval and *p* value for group difference.

## Data Availability

Not applicable.

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
