# Peer review of "Comparing the Effects of Canagliflozin vs. Glimepiride by Body Mass Index in Patients with Type 2 Diabetes and Chronic Heart Failure: A Subanalysis of the CANDLE Trial"

_biomedicines, 2022, doi:10.3390/biomedicines10071656_

Round 1

Reviewer 1 Report

Diabetes and cardiovascular diseases are the biggest problems of our civilization. All research in this direction is needed and highly desirable. The publication, or actually a series of publications of the authors on the influence of SGLT2 glucose transporter inhibitors on NT-proBNP level, raise important topics and I consider them valuable. Different parameters were taken into account by the researcher, such as  eGRF and hemoglobine, hematocrit and A1c Hb. It seems very interesting to relate the effects of the tested inhibitors to BMI (bod mass index). The subject of clinical trials in this case were two inhibitors sulfonylurea glimepiride and canaglifozin. The study lasted 24 weeks, so long enough to draw conclusions. Detailed statistical analysis of each step was performed. Tables and charts are legible, neat and transparent. Although the difference is between taking them not statistically very significant indisputably has been suggested that changes in NT-proBNP differ between patients with vs without obesity (which is manifested by the BMI index). The effect of inhibitors on body weight is very promising. The discussion is interesting because it also applies to other parameters that change after treatment with SGLT2 inhibitors, such as the level of leptin, adiponectin and moreover influence on level of insulin. This is only a confirmation that many parameters and the body as a whole should always be taken into account in the treatment of systemic diseases such as diabetes and heart disease.

Author Response

We sincerely thank you for critical review of our manuscript and insightful comments. It is not realistic to prove everything about SGLT2 inhibitor by this study alone. It is our pleasure if our manuscript is useful for future clinical studies of SGLT2 inhibitors. As you indicated, we would like to further investigate impacts of SGLT2 inhibitors on leptin, adiponectin and insulin levels.

Reviewer 2 Report

This manuscript report a subanalysis of the CANDLE trial (introduction 51-54, ref.8,9). If I´m allowed to put forward a boldly simplified conclusion, there is no difference between a SGLT2 inhibitor and a sulfonylurea between nonobese and obese patients (with type 2 diabetes and heart failure). It could mean that obesity is no factor in deciding which of these two classes of antidiabetics should be administered (beyond the psychological improvement of the weight losing obese). However, inability to find a clinically significant difference could be caused by insufficient power, e.g. relatively short observation period (24 weeks), or relatively small number of participants. These two aforementioned limitation probably could be the reason, why the evaluation is relying on proxy (laboratory) measures like NT-proBNP as only measurement related to heart failure instead on life quality/morbidity/mortality. If not that I would surely expect at least a greater battery of laboratory results.

From the methodological point of view, there is in multiple places throughout the manuscript the confusion of when there is a difference. E.g. in Introduction (line 64) there is a statement "...showed a non-statistically significant reduction...". In fact, this only means "... did not show reduction...". I know that this is often used, but in striving to higher level of scientific rigor, this is confusing and leads to inflated expectations as to the prediction what will happen for future patients taking the tested drugs. I personally recommend to mention such "differences" only in the result section, in a more neutral way, like "lower/higher apparent average value, but no statistically significant difference", in other parts of a paper I would write bluntly "no difference", and I would surely not CITE such results in such a way as in lines 64-65.

I also have a question regarding NT-proBNP levels: If obesity per se leads to decrease it seems to be difficult to use it as a marker - one can predict that the SGLT2 inhibitor treatment would lead simultaneously to weight loss (also seen in this study) - that increasing NT-proBNP, and decrease of the ventricle load - that decreasing the same. Net change would then be modest, not much different from a sulfonylurea treatment where NT-proBNP can be predicted to stay the same.

Other remarks

Introduction

Primary and secondary endpoint of CANDLE study should be mentioned shortly, otherwise it is confusing why there are no clinical parameters followed (aside of BMI which is an independent or grouping variable for this subanalysis).

results 186-191

"Although the group ratio for patients with baseline BMI <25 kg/m2 was smaller than that of patients with baseline BMI ≥25 kg/m2, there was no statistical difference by baseline BMI subgroups (P=0.222). When proportional change in NT-proBNP geometric means were modeled by continuous baseline BMI values, the magnitude of proportional change in canagliflozin group was lower than that of glimepiride group in BMI <30 kg/m2 approximately (Figure 2c)"

From figure 2c there is apparently wide region of overlap of the 95% confidence intervals for the treatments over all BMI levels, so it is not true that the change is lower (see also the remark about statistical significance above).

Figure 2c (see also the remark above) will greatly benefit if individual datapoints are depicted. Such as now we need to believe implicitly in the mathematical model used. Also other graphs can include individual data points if their clarity is not compromised.

Discussion 317-318

"Immediately after patients receive an SGLT2 inhibitor, hematocrit levels increase

due to dehydration; this transient increase in hematocrit is a result of increased erythropoietin" This sounds confusing, I guess that dehydration causes the transient effect, long term effect - after compensation of dehydration - is caused by increased erythropoietin

Author Response

We sincerely thank you for critical review of our manuscript and valuable comments. Please find below our responses. Your comments and suggestions did improve the quality of this manuscript. We would appreciate it if you could review the manuscript again.

This manuscript reports a subanalysis of the CANDLE trial (introduction 51-54, ref.8,9). If I´m allowed to put forward a boldly simplified conclusion, there is no difference between a SGLT2 inhibitor and a sulfonylurea between nonobese and obese patients (with type 2 diabetes and heart failure). It could mean that obesity is no factor in deciding which of these two classes of antidiabetics should be administered (beyond the psychological improvement of the weight losing obese). However, inability to find a clinically significant difference could be caused by insufficient power, e.g. relatively short observation period (24 weeks), or relatively small number of participants. These two aforementioned limitation probably could be the reason, why the evaluation is relying on proxy (laboratory) measures like NT-proBNP as only measurement related to heart failure instead on life quality/morbidity/mortality. If not that I would surely expect at least a greater battery of laboratory results.

→We thank you for valuable comments. As you pointed out, this study was unable to demonstrate significant differences in some parameters. This may be due to shorter observation period (24 weeks) and a small number of patients, as you suspected. We discussed these at Limitation section.

From the methodological point of view, there is in multiple places throughout the manuscript the confusion of when there is a difference. E.g. in Introduction (line 64) there is a statement "...showed a non-statistically significant reduction...". In fact, this only means "... did not show reduction...". I know that this is often used, but in striving to higher level of scientific rigor, this is confusing and leads to inflated expectations as to the prediction what will happen for future patients taking the tested drugs. I personally recommend to mention such "differences" only in the result section, in a more neutral way, like "lower/higher apparent average value, but no statistically significant difference", in other parts of a paper I would write bluntly "no difference", and I would surely not CITE such results in such a way as in lines 64-65.

→We thank you for your valuable advices. Per suggestion, we revised the manuscript to avoid any confusions among audience.

I also have a question regarding NT-proBNP levels: If obesity per se leads to decrease it seems to be difficult to use it as a marker - one can predict that the SGLT2 inhibitor treatment would lead simultaneously to weight loss (also seen in this study) - that increasing NT-proBNP, and decrease of the ventricle load - that decreasing the same. Net change would then be modest, not much different from a sulfonylurea treatment where NT-proBNP can be predicted to stay the same.

→We thank you for this very important comment. It is reported that NT-proBNP concentration is higher if a patient is lean while it is lower in an obese patient. As you may know, there are many reports about reduction of body weight and ventricular load by SGLT2 inhibitor. In this sub-analysis, the canagliflozin group resulted in a decrease in BMI, but NT-proBNP was not different between both drug groups and between before after treatment with canagliflozin. One of the possible reasons behind this phenomenon appears to be a decrease in body weight by canagliflozin and a resultant offsetting the NT-proBNP reduction mediated by the canagliflozin-induced cardiac unload. Since SGLT2 inhibitor deceases body weight, NT-proBNP might have been an inappropriate biomarker to directly reflect the reduction in cardiac load via SGLT2 inhibition. However, there are still unresolved aspects behind this phenomenon, and there is room for discussions. It is therefore necessary to search for more appropriate biomarkers for efficacy assessment and monitoring of cardiovascular effects by SGLT2 inhibitor. This is an important point, and we included this into Discussion section.

Introduction

Primary and secondary endpoint of CANDLE study should be mentioned shortly, otherwise it is confusing why there are no clinical parameters followed (aside of BMI which is an independent or grouping variable for this subanalysis).

→We thank you for sound advices. We described the endpoints of the CANDLE study at Introduction section.

results 186-191

"Although the group ratio for patients with baseline BMI <25 kg/m2 was smaller than that of patients with baseline BMI ≥25 kg/m2, there was no statistical difference by baseline BMI subgroups (P=0.222). When proportional change in NT-proBNP geometric means were modeled by continuous baseline BMI values, the magnitude of proportional change in canagliflozin group was lower than that of glimepiride group in BMI <30 kg/m2 approximately (Figure 2c)"

From figure 2c there is apparently wide region of overlap of the 95% confidence intervals for the treatments over all BMI levels, so it is not true that the change is lower (see also the remark about statistical significance above).

Figure 2c (see also the remark above) will greatly benefit if individual datapoints are depicted. Such as now we need to believe implicitly in the mathematical model used. Also other graphs can include individual data points if their clarity is not compromised.

→Your comment is correct. We revised the result in Figure 2c. During the process of revision, it was suggested that new findings might be obtained in the sub-group of BMI>30kg/m2 in the canagliflozin group. We described this as a future study hypothesis at Discussion section. In this statistical modeling, we used restricted cubic spline function which is more flexibly fitted to data than simple linear models, therefore mis-fitting to data is less likely than those in simple linear regression analyses.

Discussion 317-318

"Immediately after patients receive an SGLT2 inhibitor, hematocrit levels increase due to dehydration; this transient increase in hematocrit is a result of increased erythropoietin" This sounds confusing, I guess that dehydration causes the transient effect, long term effect - after compensation of dehydration - is caused by increased erythropoietin

→Your comment is correct. This study did not measure erythropoietic concentration. Hence, our argument was hypothetical. We revised the manuscript at Discussion section accordingly.

Reviewer 3 Report

I reviewed the manuscript titled Comparing the effects of canagliflozin vs glimepiride by body mass index in patients with type 2 diabetes and chronic heart failure: a sub-analysis of the CANDLE trial.

The study is interesting, the topic is important and the article is well written. However I did notice a few shortcomings.

1)      The authors analyzed two subgroups of BMI  >25 and <25 .  A rational for subgrouping is needed. Why 25?

2)      30 Abstract.  Was “slighty smaller increase” of NT-proBNP statistically  significant? P-value should be presented.

3)      36 Abstract   Authors divided the group on  subgroup BMI> 25 and < 25, then results were presented in the group  BMI < 30. Confusing.

4)      191, fugure 2 c, p-value should be presented. Was the result statistically significant?

5)       If there was not statistically significant difference and only trend was observed, it cannot be concluded that BMI effects NT-proBNP level.

6)      Conclusions need to be changed, authors go too far.

Author Response

We sincerely thank you for critical review of our manuscript and various valuable comments. Please find below our reply. Your review helped our manuscript revised much better. We would appreciate your review again.

  • The authors analyzed two subgroups of BMI >25 and <25. A rational for subgroupingis needed. Why 25?

→We thank you very much for valuable comment. We used BMI >25 and <25 as the threshold in this study based on the World Health Organization (WHO) of obesity, i.e., BMI >25 as obesity. Hence, we used this criterion to conduct sub-group analysis of obesity versus non-obesity. Ideally, we sought to divide obesity into I, II, and III. However, due to small ratio of obesity in Japanese and a small number of study subjects in this study, we could not conduct such detailed grouping. We discussed this at Materials and Methods and Limitation sections.

  • 30 Abstract.  Was “slightly smaller increase” of NT-proBNP statistically  significant? P-value should be presented.

→We thank you for your valuable advices. As you pointed out, we have added the P value.

  • 36 Abstract  Authors divided the group on subgroup BMI> 25 and < 25, then results were presented in the group BMI < 30. Confusing.

→Your comment is correct. The result of BMI<30 is exploratory. However, this is an intriguing result from this sub-analysis and should be investigated in a future study. At the conclusion section of Abstract, we will not comment on BMI<30.

  • 191, fugure 2 c, p-value should be presented. Was the result statistically significant?

→We thank you for your valuable advices. As you pointed out, we have added the P value in Figure 2c.

  • If there was not statistically significant difference and only trend was observed, it cannot be concluded that BMI effects NT-proBNP level.

→We thank you very much for valuable comment. Your comment is correct. In this sub-analysis, BMI did not demonstrate significant differences in NT-proBNP. Accordingly, we have toned-down and revised wording in Conclusion section.

  • Conclusions need to be changed, authors go too far.

→We thank you very much for valuable comment. We have toned-down and revised the conclusions appropriately.

Round 2

Reviewer 3 Report

The manuscript has been corrected by authors and can be published in current form.